# Human Health Risk Assessment Is Associated with the Consumption of Metal-Contaminated Groundwater around the Marituba Landfill, Amazonia, Brazil

**DOI:** 10.3390/ijerph192113865

**Published:** 2022-10-25

**Authors:** Thaís Karolina Lisboa de Queiroz, Volney de Magalhães Câmara, Karytta Sousa Naka, Lorena de Cássia dos Santos Mendes, Brenda Rodrigues Chagas, Iracina Maura de Jesus, Armando Meyer, Marcelo de Oliveira Lima

**Affiliations:** 1Programa de Pós-Graduação em Saúde Coletiva (UFRJ/IESC) 1, Universidade Federal do Rio de Janeiro (UFRJ), Rio de Janeiro 21941-901, Brazil; 2Seção de Meio Ambiente (SAAMB) 3, Instituto Evandro Chagas (IEC/SCTIE/MS), Ananindeua 67030-000, Brazil; 3Programa de Pós-Graduação em Epidemiologia e Vigilância em Saúde (PPGEVS/IEC/SCTIE/MS) 2, Instituto Evandro Chagas (IEC), Ananindeua 67030-000, Brazil

**Keywords:** Epidemiology 1, environmental exposure 2, risk assessment 3, Amazon 4, ICP-MS 5

## Abstract

Groundwater is present in its purest form beneath the earth’s surface. However, metal contamination is potentially a problem faced by many countries. For this reason, the present study aims to make an assessment of the risks associated with groundwater consumption around the Marituba landfill in an Amazon region. The present study was characterized as transversal with the use of primary data. The sampling occurred in a stratified random way, performed in two radii of action being the 1st radius of 2.5 km away from the landfill and the 2nd radius of 3.5 km away from the landfill to the neighborhoods. A total of 184 points were collected. In all communities the average daily dose (ADD) was higher than the reference oral dose (Rfd), for the metals As, Pb and Mn the risk quotient (HQ) was greater than 1 (one) in all neighborhoods, the concentration of Mn in the least exposed neighborhood was greater than 10 µg.L^−1^, even at a distance of 3.5 km from the landfill. The average concentrations for As and Pb did not exceed the recommended, however, they were more significant for the Beira Rio neighborhood, respectively 1.47 µg.L^−1^ and 1.9 µg.L^−1^. And the average concentration for Cu was more significant for the Uriboca neighborhood 18.20 µg.L^−1^, but within the recommended. The average of the general concentration of Heavy Metals Pollution Index (HPI) of the water consumed was 80.03, indicating that the water consumed by the population is contaminated by metals.

## 1. Introduction

Groundwater is present in its purest form beneath the Earth’s surface and has numerous advantages over surface water for human supply, including the fact that it is cheaper, and more abundant, with less contamination and loss through evaporation [1].

The northern region of Brazil is traditionally known for its abundance of freshwater, but also has a low socioeconomic development index, absence of an adequate infrastructure for a sanitary sewerage network, and generally distributes low-quality water for human consumption [2].

The water supply in the metropolitan region of Belém (RMB) is carried by an insufficient system in terms of structure, technology, and quantity and quality of services offered. Moreover, interruptions in the supply of drinking water to the population owing to a lack of maintenance in the water supply network are common [3].

Groundwater is considered the main source of human consumption in much of the RMB, and its use has been increasing worldwide. This statement becomes even more pertinent when considering that water consumption doubles every 20 years, and that in the last 50 years its availability per inhabitant has decreased by 60% [4].

In the City of Marituba, the groundwater is classified as shallow, most of the wells built in the region have little depth, especially the Amazon type wells. The wells built in the region for the purpose of water distribution for public supply are insufficient, and close to areas of possible contamination, such as waste disposal, gas stations and cemeteries [3].

Metal contamination is a problem faced by several countries and affects various areas, from the community to agriculture. Several factors can contribute to the presence of metals, such as fertilizer use, plant nutrients, and discharge of domestic and/or industrial effluents into water bodies [5,6] In addition to the nature and content of pollutants in the leachate, which can vary depending on the composition of the waste age of the waste, hydrology of the site, and the way in which the waste is treated in the solid waste disposal areas [7].

To quantitatively assess the potential risk, human health risk assessments are conducted to determine whether exposure to a chemical at any dose could cause an increase in the incidence or adverse effects on human health [8]. The importance of this study for the region is necessary because the areas visited are vulnerable to the risk of contamination. There is no public supply network sufficient for the distribution of water for human consumption of good quality for the RMB. It is known that this is the first study conducted in the region on risk assessment to human health, the present study aims to make an assessment of the risks associated with the consumption of groundwater through the quantification of metals As, Cu, Pb and Mn in the surroundings of the landfill of Marituba. And perform the pollution index for heavy metals (HPI).

## 2. Materials and Methods

### 2.1. Study Area

The RMB comprises the municipalities of Belém, Ananindeua, Marituba, Benevides, and Santa Bárbara, and covers an area of approximately 1200 km^2^, corresponding to 0.1% of the surface area of the State of Pará.

Marituba is the smallest municipality in the territorial extension of the State of Pará, with an area of approximately 100 km² and an estimated population of 129,321 inhabitants in 2018, with the majority residing in the urban area (99%), which is approximately 11 km from the capital [9].

The geology of the municipality of Marituba is the same as that found throughout the entire area of the Belém micro-region, represented by Tertiary sediments of the Barrier formation consisting of sandstones, siltstones and argillites, and by the unconsolidated sediments of the sub-current and recent Quaternary [3,10].

The activities were developed in the areas surrounding the landfill, located in the municipality of Marituba, in an area of approximately 1 km². The neighborhoods of São João, Santa Lúcia I, Beira Rio (in the Santa Lúcia II community), and Uriboca (in the Campina Verde community) in the first radius of action, 2.5 km away from the landfill of Marituba, were considered the most exposed populations. The neighborhoods of Decouville (in the Beija Flor community) and São Pedro, comprising the second radius of action, 3.5 km away from the landfill, were considered the less exposed populations (see Figure 1).

### 2.2. Sampling Design

The present cross-sectional study used primary data. The sampling occurred in a stratified random way, and was performed in two radii of action: the 1st radius was 2.5 km away from the landfill, and the 2nd radius was 3.5 km away from the landfill in the surrounding neighborhoods. This method was selected to contemplate the largest possible number of residences, improve the characterization of the area, and obtain different exposure points [11,12]. The study was developed with 184 residential groundwater collection points, and a structured questionnaire was applied to the neighborhoods covered by the two radii of action. Characteristics, such as the depth and type of wells found in the region and the geographical location of the households (Appendix A).

### 2.3. Groundwater Collection

The samples the underground water samples from tubular wells were collected using a bucket with nylon string because it is a closed well, and only a small opening is available for collecting, and the water samples from the Amazon type wells were collected using a stainless bucket and rope, each point was collected in duplicate and then and were transferred to a 250-mL collection flask for metals in perfluoroalkoxy (PFA), which had been previously washed with a 10% HNO_3_ solution, and rinsed with ultrapure deionised water (MILLI-Q^®^, Millipore, Advantage A10^®^) in abundance, to remove any traces of acid from the container, for subsequent use and stored, and transported in thermal boxes containing dry ice [13], until arrival at the Laboratory of Metals and Ecotoxicology, the Environmental Section of the Evandro Chagas Institute.

### 2.4. Structured Questionnaire

The present study focused on the population that consumes groundwater in RMB in the municipality of Marituba in communities located around the landfill. Through a preliminary investigation, it was possible to identify individuals who consumed water from underground sources and, therefore, to exclude those who consumed water from other sources, such as surface or mineral water.

To improve the risk assessment for individuals exposed to water consumption, structured questions were generated to obtain data from the participants, including name, address, neighborhood, age, sex, education, economic situation, exposure time, type of residence, source of drinking water, and forms of water treatment.

To provide information on the individuals’ minimum daily water consumption, we used information suggested by the Scientific Organization of the United States Institute of Medicine (IOMS), which establishes the daily consumption of water by age group (see Table 1).

Table 1 water consumption recommended by the Scientific Organization of the United States Institute of Medicine (IOMS) according to age group.

The questionnaires were administered to individuals aged ≥ 6 years and the study was submitted to the Ethics Committee on Research with Human Beings at the Evandro Chagas Institute and approved by Opinion No. 4.160.789. The informed consent form (TALE) was used for minors and the informed consent form (TCLE) was used for adults and/or legal guardians.

### 2.5. Sample Preparation

#### 2.5.1. Groundwater

An aliquot of approximately 15 mL of each sample collected was filtered using a cellulose acetate filter membrane with a porosity of 0.45 µm and 0.47 mm diameter (Millipore, Merck) through a Kitassato filtration apparatus with the aid of a vacuum pump. After filtration, the sample was transferred to a 15-mL polyethylene conical tube, which was properly identified and subsequently acidified with concentrated bidistilled nitric acid (HNO_3_) (destilacid). At the end of this process a final concentration of 1% acid (*v*/*v*) is obtained in the sample.

#### 2.5.2. Metal Analysis

The concentrations of metals, such as As, Cu, Pb, and Mn, in the collected samples were quantified using an inductively coupled plasma mass spectrometer (ICP-MS) BRUKER model 820-MS. All samples were analyzed in a clean room environment, class 1000, in the Laboratory of Metals and Ecotoxicology, of the SEAMB of the Evandro Chagas Institute (IEC), located in the city of Ananindeua, Pará State.

During the analyses, the analytical calibration curves were prepared and calibrated with 1% (*v*/*v*) HNO_3_ solution, simulating the conditions of the samples after the initial treatment. For analytical quality control, we performed readings of certified reference materials for water analysis (1640a Trace Elements in Natural Water, obtaining the following recoveries for the analytes As (102.40%), Cu (96.29%), Pb (97.07%) and Mn (97.06%) and 1643f Trace Elements in Water I, National Institute of Standards and Technology [NIST] mark), with recovery for As (112.41%), Cu (107.71%), Pb (96.94%) and Mn (101.08%).

#### 2.5.3. Statistical Analysis

Data were statistically analyzed using Microsoft Excel 2013 and Epi InfoTM version 7.2.5 of 2021 to calculate the frequency of responses to the variables in question. Descriptive statistics and measures of central tendency (mean), median, and dispersion (standard deviation) were obtained. Minitab^®^ software (version 19.0) was used to perform association analysis, either parametric or non-parametric according to the type of variable analyzed, to verify the statistical significance of the results.

The general calculation shown in Equation (1) was used to identify the estimates of the magnitude, frequency, and duration of human exposure to each potentially toxic metal in the environment, which are normally reported as the average daily dose (ADD) [14,15].

Risk characterization is the final step in health risk assessment. The health risk of groundwater consumption was assessed in relation to its chronic (non-carcinogenic) effects based on the calculation of ADD estimates and toxicity values defined for each potentially toxic metal according to the following equations: The non-carcinogenic risk was calculated as the hazard quotient (HQ) as shown in Equation (2). To calculate the Heavy Metal Pollution Index (HPI), it involved the steps described in Equation (3).

Equation (1). Calculation of the estimated magnitude, frequency, and duration of exposure.
(1)ADD=(C × IR × EF×ED)/(BW ×AT)
where ADD indicates the duration of exposure (mg/kg-day); C indicates the concentration of the substance in the contact medium (e.g., mg/L of water); IR indicates the ingestion rate (L/day); EF indicates the exposure frequency (days/year); ED indicates the duration of exposure (year); BW indicates the body mass (kg); and AT indicates the average time (day);

Equation (2). Calculation of the Risk Quotient.
(2)$$\mathrm{HQ}=\frac{\mathrm{ADD}}{\mathrm{RfD}}$$
where HQ indicates the risk quotient; ADD indicates the average daily dose; and RfD indicates the reference dose.

Equation (3). Heavy metal pollution index (HPI).

The Calculation for HPI involved the following steps:(a)First, the calculation of the ith parameter weighting:
(3a)$$\mathrm{Wi}=\frac{K}{\mathrm{Si}}$$
where Wi: Weighing unit; Si: the recommended standard for the ith parameter; K: Proportionality constant.(b)Second, the calculation of the quality rating for each of the metals:
(3b)$$\mathrm{Qi}=\frac{100\times \mathrm{Vi}}{\mathrm{Si}}$$where Qi: is the subindex of the ith parameter; Vi: is the monitored value of the ith parameter; Si: is the standard or allowable limit for the ith parameter.(c)Third, the sum of these sub-indices into the overall index. Calculating the HPI:
(3c)$$\mathrm{HPI}=\frac{\mathrm{Qi}\times \mathrm{Wi}}{1}$$where Qi: is the sub-index of the parameter ith; Wi: is the weighing unit for the parameter ith; The critical value of the pollution index is 100. For the present study; Si: value was taken from World Health Organization (WHO), except only for Mn, due to absence of tolerable limit by WHO, CONAMA Res. 460/2013 was used for Mn.

### 2.6. Assessment of Potential Risk to Human Health

The evaluation of potential risk to human health in a screening approach was performed based on the procedures described in the USEPA [14]. In this approach, conservative values of exposure and risks to human health are assumed, with the aim to prioritize non-hazardous contaminants in a worst-case scenario and to select those that deserve further study (e.g., a higher sampling frequency).

Risk assessment is a function of hazard and exposure, and is defined as the process of estimating the probability of occurrence of any likely magnitude of adverse health effects over a specified period. The toxicity indices for each potentially toxic metal are listed in Table 2 [16].

The existence of a risk to groundwater intake will be confirmed when the concentration of the chemicals of interest in the samples is more than that recommended by the current legislation.

The input parameters for the ADD formulas are listed in Table 3. The non-cancer risk was calculated as the hazard quotient (HQ) as shown in Equation (2) above. An HQ ≤ 1 indicates an acceptable risk level, while an HQ > 1 represents an unacceptable risk of non-cancer effects [17].

## 3. Results

### 3.1. Concentrations of Metals in the Groundwater

Table 4 shows the mean and standard deviation for the concentrations of dissolved metals and metalloids for As, Cu, Pb, and Mn in groundwater samples consumed by the population of the following neighborhoods: São João, Beira Rio (in the Santa Lúcia II community), Santa Lúcia I, Uriboca (in the Campina Verde community), Decouville (in the Beija-flor community), and São Pedro, located around the Marituba landfill. To compare the average groundwater samples, the results were compared with the drinking groundwater standards of the US Environmental Protection Agency [18], World Health Organization [19], Portaria GM/MS n° 888/2021, and the Resolution of the National Environment Council (CONAMA) n° 460/2013.

### 3.2. Human Health Risk Assessment

Table 5 shows the mean values calculated for the risk quotient and the average daily dose in the groundwater of the neighborhoods surrounding the landfill. For the HQ results, in all neighborhoods it was higher >1, indicating a possible exposure of this population to non-carcinogenic risks for drinking water. While the results for ADD indicate that the water consumed by the resident population is unfit for human consumption in relation to metals.

### 3.3. Heavy Metal Pollution Index (HPI)

The HPI were evaluated using the average concentrations of metals quantified in groundwater samples from artesian and Amazon-type wells in the region surrounding the landfill. The average results of the pollution indices for heavy metals are presented (Appendix A).

The average of the general concentration of the Heavy Metal Pollution Index of the water consumed by the population living in the neighbourhoods Beira Rio, Decouville, Santa Lúcia I, São João, São Pedro and Uriboca was 80.03, being classified between 76 and 100, considered as very poor water, as shown in Table 6. Consequently, the result of this HPI value indicates that the water consumed by the population of the neighborhoods, are contaminated by metals.

## 4. Discussion

### 4.1. Concentrations of Metals in the Groundwater

#### 4.1.1. Arsenic (As)

The mean concentrations of dissolved arsenic in the groundwater samples for all of the included communities did not exceed the value established by the international drinking water standards, such as those of the WHO and USEPA of 10 µg.L^−1^, and values established in Brazil, such as the CONAMA resolution n° 460/2013 of 10 µg.L^−1^ and Portaria n° 888, which suggests 1000 µg.L^−1^. Concentrations of As < 10 mg.L^−1^ are due to the non-occurrence of groundwater contact with rock or soil, which have primary or secondary metal minerals [21].

Exposure to As occurs mainly through the ingestion of drinking water or food. Given that As is odorless and colorless, people are commonly unaware that they are absorbing this invisible toxins [22]. In this study, exposure to As was considered through the use of groundwater for consumption by the human study population. In developing countries, such as Brazil, mainly to the north of the Amazon, groundwater from the Amazon or tubular-type wells are common for potability, and are often the only means to obtain water for human consumption [23].

The mean concentrations of As in groundwater in the neighborhoods located around the landfill site were lower than those found in groundwater wells in other parts of the world, such as around landfill sites in Bangladesh City, which ranged from <0.5 to 3.200 µg.L^−1^, where geological features and the environment, contribute to the mobilization of the metal into the wells [24]. Moreover, in the surroundings of the Tibetan landfill, the average concentration of As in groundwater was found to be 0.5 µg.L^−1^, which was similar to the average found for Decouville 0.49 µg.L^−1^ in the present study [25].

#### 4.1.2. Copper (Cu)

The mean dissolved Cu concentrations were within the US Environmental Protection Agency (USEPA) standards of 1300 µg.L^−1^. Among the included communities, Uriboca had the highest average Cu concentration of 18.20 µg.L^−1^, which may be associated with its proximity to the Marituba landfill.

Cu, due to its low solubility, is generally present in groundwater at concentrations < 1 µg.L^−1^ [13]. Although Cu is considered an essential element for all forms of life at low concentrations, at high concentrations, it is associated with gastrointestinal problems, such as diarrhea, abdominal pain, nausea, and vomiting [26].

The concentrations of Cu found in the groundwater around the Marituba landfill site were similar to those found in the groundwater near the landfill in the city of Ribeirão Preto, Brazil, where the highest observed concentration was 20.90 µg.L^−1^ and the lowest was 1.39 µg.L^−1^ [27,28]. However, in the Federal District in Brasilia, the Jockey Landfill had an average Cu concentration of 100 µg.L^−1^, which was much higher than that in the present study. The Cu concentrations in the present study may have been influenced by the type of well, as well as its depth, and the proximity of the neighborhood to the landfill site [29].

The mean concentration of Cu found in groundwater in other countries, such as the Kumasi region around the Oti Landfill was 247 µg.L^−1^, which is much higher than the concentrations found in the present study [30]. Moreover, in the surrounding areas of the Vientiane landfill site in Laos, the mean Cu concentration in groundwater samples was 10 µg.L^−1^ lower than that of the present study for the Uriboca community (18.20 µg.L^−1^) [31].

#### 4.1.3. Lead (Pb)

The mean concentrations of dissolved Pb in the groundwater samples in all of the studied neighborhoods were within the international and national quality standards for drinking water. Among the neighborhoods, Beira Rio (in the Santa Lúcia II community) obtained the highest average Pb concentration of 1.9 µg.L^−1^, which may be due to it being one of the closest neighborhoods to the sanitary landfill.

In other Brazilian Cities, as in São Paulo, the average concentration of Pb in groundwater around the Ribeirão Preto landfill was 3.78 mg.L^−1^ [28,29], which was higher than that found in the present study. The average concentration found in the surroundings of the landfill known as the Jockey Landfill, mentioned above, was 273 µg.L^−1^. Therefore, it is inferred that a higher average concentration of Pb is related to the altitude at which the wells were constructed: the lower the terrain, the greater the chances of leachate percolation into the water table to reach the groundwater, the type and depth of the wells constructed, and the more acidic the pH conditions [29].

The concentration of Pb found in the northern Amazon region in the present study was close to the average concentration found in groundwater in other parts of the world, such as in southern Norway around the Revdalen Landfill, with an average of 2.4 µg.L^−1^. It is worth noting that landfills in Southern Norway have a lifespan of 21 years [7], while the Marituba landfill has a life span of only 8 years, with the full operation starting in 2015.

In contrast, in the Oti Landfill in the Kumasi region, the average concentration of Pb in groundwater was 94 µg.L^−1^, which exceeded the limits recommended by the USEPA and WHO, and was higher than the averages for Pb in the present study [30]. It is known that different types of wells can interfere with the high concentrations of metals in groundwater, which is similar to the conditions observed in the present study, in which the wells were Amazon type (wells with an open mouth and generally without structures in their internal lateral constructions) and tubular type (with a closed mouth and generally concreted in their internal structures).

The average concentration of Pb in groundwater around the Vientiane landfill site in Laos was 1.7 µg.L^−1^, which was approaching the concentration of Pb recorded in the Beira Rio community of 1.96 µg.L^−1^ in this study [31].

#### 4.1.4. Manganese (Mn)

The average concentrations of dissolved manganese in the waters of the neighborhoods of Beira Rio (in the Santa Lúcia II community) and Santa Lúcia I, located very close to the landfill, were >10 µg.L^−1^, which is the limit recommended in CONAMA Resolution No. 460/2013. In the community of Santa Lucia I, the concentration of Mn was 17.79 µg.L^−1^, while that in Rio was 20.65 µg.L^−1^. However, the Decouville neighborhood showed the highest concentration of Mn, reaching 37.77 µg.L^−1^.

The present study presents lower average concentrations of Mn in the underground waters than in other regions of Brazil. For example, in São Paulo, in the sanitary landfill of Ribeirão Preto, the concentration of Mn found in the underground water samples was 74.381 mg.L^−1^ higher than the average found for the communities in the present study [28].

The mean concentrations of Mn found in groundwater in other parts of the world were higher than those found in the present study. In the vicinity of the Zhoukou landfill site in China, the mean concentrations of Mn exceeded the limit of 100 µg.L^−1^ [32,33]. Moreover, the mean concentrations of Mn in the groundwater in the surroundings of the Lahijan landfill, located in northern Guilan Province, was 1.77 mg.L^−1^ [34]. In the surroundings of the Sichuan landfill site in one province in Southwest China, the mean Mn concentration in the groundwater was 1.06 mg.L^−1^ [17]. Moreover, the mean Mn concentration found in Cipayung landfill in Depok, Indonesia was 0.84 mg.L^−1^, which was well below the levels found in this study [11].

Studies have shown that the Mn concentrations in groundwater decrease with increasing distance of wells from landfill areas. According to Erlinna [35], the decrease in Mn concentration starts at a distance of 150 m at 0.1 mg.L^−1^, while at a distance of >150–250 m, a constant result of 0.2 mg.L^−1^ is obtained. However, in the present study, the Mn concentration in the Decouville neighborhood (in the hummingbird community) was the highest among all of the measured averages, despite being one of the neighborhoods located farthest from the landfill by approximately 1584 m, where the population is less exposed [11,32,33].

### 4.2. Human Health Risk Assessment

#### 4.2.1. Arsenic (As)

Based on the results of the calculations, Table 5 shows that the average HQ concentrations for As in all neighborhoods studied were greater than those established for HQ > 1. This indicates that the population may present unacceptable non-carcinogenic health risks and that the underground water consumed by the resident population has a potential non-carcinogenic risk to health. The highest levels of HQs were found in the Beira Rio neighborhood (in the community Santa Lúcia II) at 2.247 × 10^4^, and São João at 3.524 × 10^4^, both of which are located in the first radius of action and close to the sanitary landfill. Moreover, in the São Pedro neighborhood, one of the neighborhoods located in the second radius of action, the HQ was 3.466 × 10^4^, and was considered a possible control neighborhood of the present study.

In a study conducted in the groundwater around the landfill in Chandigarh, India, The HQ levels for As of 136 were higher than those found in the present study in the region of Marituba [36]. However, in the vicinity of the landfill in the city of Sialkot, in the province of Punjab-Pakistan, the mean HQ value for As in groundwater was 1.047 × 10^−6^ lower than that of the Marituba region [37].

The results for ADD were higher than the oral As intake reference dose (RfD) recommended by [18] of 3 × 10^−4^ mg.kg^−1^/day, as shown in Table 1. The values found for each neighborhood were as follows: São João, 1.058 × 10^1^; Uriboca, in the Campina Verde community, 7.800; Santa Lúcia I, 8.754; Beira Rio, in the Santa Lúcia II community, 1.274 × 10^1^; Decouville, in the Beija-flor community, 4.247; and São Pedro, 1.040 × 10^1^. All of the results are expressed as mg.kg^−1^/day. The results showed the possibility of adverse effects on human health, indicating that the water consumed by the resident population is unfit for human consumption in relation to the determination of metals and metalloids.

A study conducted on the groundwater around the landfill site in the city of Sialkot in the province of Punjab-Pakistan, the average obtained for ADD was 3.142 × 10^−10^ mg.kg^−1^/day [37], while higher levels were found in the present study.

#### 4.2.2. Copper (Cu)

Table 5 shows that the São João and Uriboca neighborhoods presented the highest results of HQ, with values of 1640.167 × 10^3^ and 3943.334 × 10^3^, respectively. These neighborhoods are located closer to the landfill, which have economies that are focused on the development of production and marketing activities of vegetables and the cultivation and sale of ornamental plants. Therefore, water is used not only for human consumption, but also for cooking, washing food, irrigation of vegetable gardens, and other routine activities undertaken by the community.

In one of the neighborhoods located in the second radius of action, Decouville, in the Beija-flor community, the average HQ concentration was higher than the established unit HQ > 1. This indicates that the water consumed by the resident population of the neighborhood may have effects that are not carcinogenic but may still be harmful to human health, thereby being considered unsuitable for human consumption in relation to the determination of metals addressed in this study. In the other neighborhoods, the results were HQ ≤ 1, indicating that the consumed water did not present risks to humans.

A study conducted in Brazil, around the controlled landfill site of Morretes, in the region of Paraná, showed that the HQ concentration for Cu was 4.6 × 10^−2^, which took into account groundwater as an exposure route through the ingestion of water by the individual. Likewise, the scenario presented in the present study presented an average HQ that was higher than that found in the region of Morretes [38].

Similar results were found in other regions of the world around the landfill of Vehari, where the average concentration of HQ was 0.233, which was lower than that found for Cu in the region of Marituba [39]. In another region, in the surroundings of the Chandigarh landfill in India, Cu HQ levels of 8.1 were found in groundwater [36]. Moreover, in the region of India, in Mohali City, the HQ for Cu was 0.01, which was well below the average values found for HQ in the neighborhoods of Marituba, surrounding the landfill site [40]. A similar study was conducted in Sialkot city in Punjab-Pakistan province, in which the mean HQ value was 2.592 × 10^−8^ [37].

Table 5 shows that the average results for ADD in the neighborhoods located in the first radius of action, with a distance of 2.5 km from the landfill, were as follows: São João, 6.560 × 10^1^; Uriboca (in the Campina Verde community), 1.577 × 10^2^; Santa Lúcia I, 2.097 × 10^1^; and Beira Rio (in the Santa-lúcia II community), 2.409 × 10^1^, with all results expressed in mg.kg^−1^/day. Furthermore, the results of ADD greater than the RfD recommended Cu intake for the neighborhoods located in the second radius of action with a distance of 3.5 km from the landfill, were as follows: Decouville (in the hummingbird community) 5.806 × 10^1^; and São Pedro 2.149 × 10^1^ mg.kg^−1^/day. All of these results were higher than the RfD of 4 × 10^−2^ recommended by [16] for Cu intake in the human body.

Studies in other countries showed that in wells in India, in the City of Mohali, the ADD was 5 × 10^−4^ mg.kg^−1^/day, which was lower than the average found for the region of Marituba [40], indicating that the intake of Cu through groundwater in Marituba is greater, mainly because it is considered as an essential metal for humans. However, as previously cited, when ingested at high concentrations, it can be harmful to human health, as the daily intake of Cu is 0.9–2.7 mg.L^−1^/day [19].

In the city of Punjab-Pakistan province, the average value for ADD found in groundwater was 1.037 × 10^−9^ mg.kg^−1^/day [37]. In Ogun State, Nigeria, the mean value for ADD was 1.19 × 10^−3^ mg.kg^−1^/day in groundwater from residential wells, which was much lower than the mean values found for the Marituba region [41].

#### 4.2.3. Lead (Pb)

Table 5 shows that the results found for Pb presented an HQ elevation > 1 for all neighborhoods, including São João (1.510 × 10^4^), Uriboca (2.451 × 10^4^), Beira Rio (4.854 × 10^4^), and Santa Lúcia I (1.139 × 10^4^). This was also clear in the districts farthest from the sanitary landfill, such as in Decouville (7.429 × 10^3^) and São Pedro (1.213 × 10^4^), which were approximately 3.5 km from the landfill. These results indicate that the water consumed in these locations are unfit for human consumption in relation to the determination of metals, with impacts not only on carcinogenesis, but also other aspects of human health.

The HQ values found in other parts of the world, for example, in the surroundings of the landfill of Vehari, Pakistan, the average Pb HQ concentration was 10.32 lower than that found for the municipality of Marituba [39]. In the surroundings of the Chandigarh landfill site in India, a mean Pb HQ of 19 was obtained for groundwater [36].

A similar study in another region of India, in the city of Mohali, found an HQ for Pb of 0.28, which was much lower than that found in Marituba. Although India is a densely populated region with sanitation problems, it still presented a lower HQ value than that found in the same type of region in the Amazon [40]. In Sialkot city in the Punjab-Pakistan province, the average Pb HQ value in groundwater in the vicinity of the landfill was 1.068 × 10^−9^ [37]. In the Iran region, in the Neyshabur landfill site, the Pb HQ levels in groundwater were 60.10 × 10^−7^ [42]. In Ogun State, Nigeria, the mean value for HQ was found 5.10 × 10^−2^ in groundwater from residential wells, which is quite different from the mean value found in the neighborhoods of the Marituba region [41].

Table 5 shows that the mean ADD values in the neighborhoods were as follows: São João, 5.290; Uriboca (in the Campina Verde community), 8.580; Santa Lúcia I, 3.99; Beira Rio (in the Santa Lúcia II community), 1.698 × 10^1^; Decouville, (in the Beija-flor community), 2.600; and São Pedro, 4.250, all expressed as mg.kg^−1^/day. All of these results are above the RfD recommended for Pb intake of 3.4 × 10^−4^ mg.kg^−1^/day [16], indicating that intake of Pb through water, occurs in several neighborhoods around the Marituba landfill and is independent of the distance from the constructed wells.

Similar studies have also been conducted in other countries. In the city of Mohali, an average value of 0.001 mg.kg^−1^/day was found for ADD in the groundwater around the landfill, which is lower than the average found in the neighborhoods of the Marituba region [40]. In the region surrounding the landfill in the city of Sialkot in the province of Punjab, Pakistan, the average value for ADD found in groundwater was 1.068 × 10^−9^ mg.kg^−1^/day [37]. In Ogun State, Nigeria, the mean ADD value in groundwater from residential wells was 1.53 × 10^−2^ mg.kg^−1^/day [41].

#### 4.2.4. Manganese (Mn)

Table 5 shows that the HQ results for all neighborhoods in this study were > 1. The highest HQ results were observed in the neighborhoods located closer to the landfill of Marituba; for example, in the neighborhoods of São João (5.429 × 10^3^) and Uriboca (in the Campina Verde community; 4.469 × 10^3^), as well as in the neighborhoods where the population is less exposed, approximately 3.5 km from the landfill, where the HQ value was 4.835 × 10^3^ for the São Pedro neighborhood.

Mn is present in various environmental components, including air, soil, and water. Once present in the environment, Mn cannot be decomposed and can only change its form to be bound or separated from the particles. In a study conducted around the Cipayung landfill site in Depok, Indonesia, the average HQ for Mn in groundwater was 5 × 10^−2^, within the limit set for HQ ≤ 1. However, this was lower than the average HQ for Mn found in this study, indicating that the waters surrounding the Marituba landfill may have negative and carcinogenic effects on the health of the population that consumes the water [11].

In the vicinity of the Vehari landfill site in Pakistan, the average HQ was 10.32, which was higher than that found for Mn in Marituba [39]. However, in the Jashore district in a rural region in southwestern Bangladesh, the groundwater surrounding the landfill site showed a mean HQ value of 0.348, which was lower than that found for the Marituba region [43].

Table 5 shows that the mean results for ADD in all neighborhoods were higher than the RfD recommended Mn intake of 1.0 × 10^−2^ mg.kg^−1^/day. The average values found for each of the neighborhoods were as follows: São João, 7.600 × 10^1^; Uriboca (in the Campina Verde community), 6.257 × 10^1^; Santa Lúcia, I 1.541 × 10^2^; Beira Rio (in the Santa Lúcia II community), 1.789 × 10^2^; Decouville (in the Beija-Flor community, 3.273 × 10^2^); and São Pedro, 6.768 × 10^1^, all of which are expressed as mg.kg^−1^/day.

It is noteworthy that the highest average ADD was found in the Decouville neighborhood, which was approximately 3.5 km from the landfill and considered as a less exposed population in this study. In Decouville, most of the wells built are of the tubular type (called artesian wells in the region), which are characterized by having a more appropriate physical structure, are usually closed, use pumps, and have a greater depth.

A similar study conducted in Ogun State, Nigeria, found an average ADD Mn value of 3.37 × 10^−3^ mg.kg^−1^/day in groundwater from residential wells, which was much lower than the average values found for the Marituba region [41].

### 4.3. Heavy Metal Pollution Index (HPI)

Appendix A, shows the average concentrations for As, Cu, Pb and Mn distributed in the neighbourhoods surrounding the landfill. The highest average As and Pb found for HPI were found in Decouville neighbourhood 204.08 and 333.33 respectively. But in all the studied neighbourhoods, the HPI averages for Pb were classified as improper. For Cu, the highest mean HPI was found in Santa Lucia I 41.32 and for Mn the highest mean was 13.85 for Uriboca neighbourhood. With the exception of Cu and Mn, As and Pb had averages higher than the critical limit of classification > 100, consequently considered inadequate for use in daily domestic activities and for human consumption. The mean HPI values of this study were lower than the HPI value of 518.55 [44].

And a study conducted on hand dug wells of the Ejisu-Juaben Municipality in Ghana, HPI values ranging from 319.20 to 688.05 were found, which were all above the critical value of 100 [30]. In the present study, the highest mean HPI for Pb was in Decouville neighbourhood, characterized as one of the least exposed neighbourhoods and most of the wells, classified as artesian, that is, wells that are not hand dug, are structured and of greater depth. However, it showed higher average results for HPI.

In a study conducted on groundwater around the landfill site located in Kumasi, Ghana. Averages for HPI 366.01 were found. indicated that the water sources were above the critical limit for drinking water (HPI > 100) [45]. However, in the present study the overall mean found for HPI among the neighbourhoods 80.03. being lower than the overall mean found in Kumasi, for the groundwater samples from wells.

## 5. Conclusions

In the present study, we sought to estimate the HQ and ADD for metals and metalloids (As, Cu, Pb, and Mn) in groundwater used for human consumption by the population of six neighborhoods located around the Marituba landfill site.

The results showed that the average concentrations of As, Cu, and Pb were within the national and international standards recommended for drinking water use. Only the average concentrations of Mn for the neighborhoods Beira Rio, Santa Lúcia I, and Decouville showed higher averages than the current legislation recommends. This indicates that although Mn is considered an essential metal for the human body at low concentrations, it can cause health problems when consumed at high concentrations.

The distribution of the non-carcinogenic risk was related to the average values found for the concentrations of As, which had HQ values > 1 in all neighborhoods. For the mean Cu HQ values, the neighborhoods with the highest concentrations were São João and Uriboca (in the Campina Verde community), both of which are located very close to the landfill of Marituba, distributed in the first radius of action. The mean values of Pb and Mn were higher than the recommended HQ values of ≤1. These results indicate that the population may present unacceptable non-carcinogenic health risks; thus, groundwater consumed by the resident population has a potential non-carcinogenic risk to human health.

The average daily dose for the average concentrations of As, Cu, Pb, and Mn in all neighborhoods was higher than the Rfd recommended by [16], as described in Table 2. This finding indicates that the consumption of metals through groundwater in Marituba is intense and can travel long distances through the water table.

Finally, an average Mn value higher than the limit recommended for underground water was found in Decouville, where the population is less exposed because it is located farther from the sanitary landfill. In addition, the average daily dose and the high-risk quotient for the communities Decouville and São Pedro both comprise populations that are further away from the landfill, thereby having reduced exposure.

The overall HPI calculated based on the average concentration of the metals As, Cu, Pb and Mn was calculated to be 80.03, ranked between 76 to 100 and considered as very poor. Indicating that the selected groundwater samples, were contaminated with metals. Mainly Pb contamination, where all the neighbourhoods were classified as an unfit water according to the classification in Table 6. The average As and Pb concentrations for HPI in the Decouville neighborhood indicated that the water sources were above the critical limit for drinking water (HPI > 100).

## Figures and Tables

**Figure 1 ijerph-19-13865-f001:**
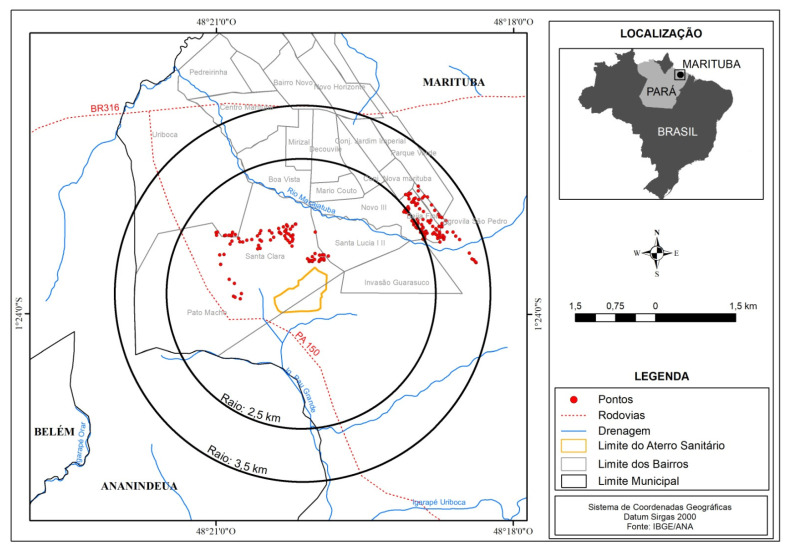
Location map of the points collected around the landfill in the municipality of Marituba, Pará State, Brazil.

**Table 1 ijerph-19-13865-t001:** Minimum water consumption by age group.

Age Group	Quantity Water (Minimum)
0–6 months	Breast milk only
7–12 months	800 mL to 1 L
1–3 years	1.3 L
4–8 years	1.7 L
9–13 years	2.1 L */2.4 L **
14–18 years	3.3 L **/2.3 L *
19–70 years	3.7 L **/2.7 L */3 L **/3.8 L ****

* Woman; ** man; **** lactating woman.

**Table 2 ijerph-19-13865-t002:** Toxicity responses (dose response) to metals as the oral reference dose (RfD) and oral slope factor (SF).

Metals	Oral RfD (mg/kg/Day)	Oral SF (mg/kg/Day)^−1^
As	3 × 10^−4^	1.50
Cu	4 × 10^−2^	n.d.
Pb	3.5 × 10^−4^	n.d.
Mn *	1.0 × 10^−2^	n.d.

n.d.: Not determined. * Mn: USEPA IRIS, 2011; Technical Guidelines, 2014; RAIS, 2017.

**Table 3 ijerph-19-13865-t003:** Input parameters to characterize the ADD value (exposure parameter).

Parameters	Description	Unit	Value
C	Concentration of metal in water	µg/L^−1^	-
IR	Ingestion rate per unit of time	l/dia	2.8 ± 0.57
EF	Frequency of exposure	Days/years	365
ED	Duration of exposure	Years	13
BW	Body weight	kg	60 ± 12
AT	Average time	Days	25.55

**Table 4 ijerph-19-13865-t004:** Concentration of metals (µg.L^−1^) in the groundwater of communities in Marituba, Pará.

Metals	Neighborhood	Average ± DP	N	USEPA 2012	WHO 2017	Port. 888/2021	CONAMA Res. 460/2013
As	Beira Rio	1.47 ± 0.77	25	10	10	1000	10
	Decouville	0.49 ± 0.26	75				
	Santa Lúcia I	1.01 ± 0.54	21				
	São João	1.22 ± 0.72	22				
	São Pedro	1.20 ± 1.02	22				
	Uriboca	0.90 ± 0.64	20				
Cu	Beira Rio	2.78 ± 4.73	25	1300	2000	2.000	2000
	Decouville	6.70 ± 22.45	75				
	Santa Lúcia I	2.42 ± 3.06	21				
	São João	7.57 ± 26.28	22				
	São Pedro	2.48 ± 5.60	22				
	Uriboca	18.20 ± 71.70	20				
Pb	Beira Rio	1.96 ± 6.89	25	15	10	1000	400
	Decouville	0.30 ± 0.68	75				
	Santa Lúcia I	0.46 ± 0.38	21				
	São João	0.61 ± 0.67	22				
	São Pedro	0.49 ± 0.40	22				
	Uriboca	0.99 ± 2.11	20				
Mn	Beira Rio	20.65 ± 38.27	25	-	-	-	10
	Decouville	37.77 ± 64.26	75				
	Santa Lúcia I	17.79 ± 11.29	21				
	São João	8.77 ± 8.01	22				
	São Pedro	7.81 ± 10.87	22				
	Uriboca	7.22 ± 9.88	20				

**Table 5 ijerph-19-13865-t005:** Non-carcinogenic risk, mean values (mg.kg^−1^/day), risk quotient (HQ) and average daily dose (ADD) in the groundwater of Marituba, Amazonia, Brazil, 2019.

	*n* = 184 Homes							
Neighborhood	HQ				ADD			
	As	Cu	Pb	Mn	As	Cu	Pb	Mn
São João	3.525 × 10^4^	1640.167 × 10^3^	1.510 × 10^4^	5.429 × 10^3^	1.058 × 10^1^	6.560 × 10^1^	5.290	7.600 × 10^1^
Uriboca	2.600 × 10^4^	3943.334 × 10^3^	2.451 × 10^4^	4.469 × 10^3^	7.800	1.577 × 10^2^	8.580	6.257 × 10^1^
Santa Lúcia I	2.918 × 10^4^	524.334 × 10^2^	1.139 × 10^4^	1.102 × 10^4^	8.754	2.097 × 10^1^	3.990	1.541 × 10^2^
Beira Rio	4.247 × 10^4^	602.334 × 10^2^	4.854 × 10^4^	1.279 × 10^4^	1.274 × 10^1^	2.409 × 10^1^	1.698 × 10^1^	1.789 × 10^2^
Decouville	1.416 × 10^4^	1451.667 × 10^3^	7.429 × 10^3^	2.339 × 10^4^	4.247	5.806 × 10^1^	2.600	3.273 × 10^2^
São Pedro	3.467 × 10^4^	537.334 × 10^2^	1.213 × 10^4^	4.835 × 10^3^	1.040 × 10^1^	2.149 × 10^1^	4.250	6.768 × 10^1^

**Table 6 ijerph-19-13865-t006:** Water quality ranking using pollution index for heavy metals (HPI). Source Elumalai, Brindha, Lakshmanan (2017) [20].

HPI Range	Quality
<25	Excellent
26 to 50	Good
51 to 75	Poor
76 to 100	Very Poor
>100	Inadequate

## Data Availability

Not applicable.

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
