# Peer review of "Human Health Risk Assessment Is Associated with the Consumption of Metal-Contaminated Groundwater around the Marituba Landfill, Amazonia, Brazil"

_ijerph, 2022, doi:10.3390/ijerph192113865_

Round 1
Reviewer 1 Report
This manuscript aims to assess the risks associated with the consumption of groundwater in the surroundings of Marituba Landfill, Amazonia, Brazil. The manuscript reads well and provide metal exposure risk to health in the communities living close to landfill, and thus can be of significance to inform the public and influence public health policy.
Some comments as follows:
· The abstract needs to add a summary of the water quality results of the study.
· Page 3, Line 116-117: what are the recoveries of the metals in the certified reference materials?
· Page 4, Line 126: add a title for the human health risk assessment section and move the Statistical Analysis section at the end.
· How is the adopted water consumption from the US (Table 1) with different climate and behavior can be used in Brazil? No such data available for the study region, or in Brazil?
· Line 131: correct the abbreviation of the average daily dose (ADI?).
· The numbers including the power of 10 must be properly written throughout the manuscript including reducing significant figures to the nearest number after the point.
Reviewer 2 Report
1. Add the key finding in abstract. it is not informative
2. Add more literature and recent papers related to human health risk
3. Add more description about recent status groundwater quality in the study area, why the study is significant in this region? What is the novelty of the present study? Add all these information in introduction section.
4. Add the specific objective of the present study at the end of the introduction
5. Sample collection techniques, methods need to detailed in the methodology section
6. I didn't find any statistical descriptive for each sampling station
7. Add the latitude and longitude of sampling site
8. I suggest to add spatial analysis for better representation of contamination site
9. Only health risk assessment was carried out in the study, i think it is not enough to publish a paper. Add pollution index, PERI, Geo-accumulation index, etc.,
10. Improve the result and discussion part, in way of find the reason or reaction or possible way of contamination in the study area
11. Improve the conclusion to satisfy the international readers
Reviewer 3 Report
This is a great paper, well structured that presents results of great interest for the research.
However, it should suffer several corrections:
lines 39 - 41 - but not only those examples (may be found any other examples with higher impact)
lines 67 - 75 - not enough details regarding groundwater depth, the stratigraphy of the area, type of soil
lines 103 - the methodology used for sample collection and preservation
line 110 - for future investigations I'll recommend using AAS rather than ICP - MS
The conclusions should be more detailed.
